# Rare protein-altering variants in *ANGPTL7* lower intraocular pressure and protect against glaucoma

Yosuke Tanigawa[1], Michael Wainberg[1], Juha Karjalainen[2,3,4], Tuomo Kiiskinen[4,5], Guhan Venkataraman[1], Susanna Lemmelä[4,5], Joni A. Turunen[6,7], Robert R. Graham[8], Aki S. Havulinna[4,5], Markus Perola[5], Aarno Palotie[2,3,4], FinnGen[¶], Mark J. Daly[2,3,4☯]*, Manuel A. Rivas[1☯]*

1 Department of Biomedical Data Science, School of Medicine, Stanford University, Stanford, California, United States of America, 2 Program in Medical and Population Genetics and Stanley Center for Psychiatric Research, Broad Institute of Harvard and MIT, Cambridge, Massachusetts, United States of America, 3 Analytic and Translational Genetics Unit, Massachusetts General Hospital, Boston, Massachusetts, United States of America, 4 Institute for Molecular Medicine Finland (FIMM), University of Helsinki, Helsinki, Finland, 5 Finnish Institute for Health and Welfare, Helsinki, Finland, 6 Department of Ophthalmology, University of Helsinki and Helsinki University Hospital, Helsinki, Finland, 7 Folkhälsan Research Center, Biomedicum Helsinki, Helsinki, Finland, 8 Maze Therapeutics, South San Francisco, California, United States of America

☯ These authors contributed equally to this work.
¶ FinnGen members are listed in S1 Text.
* mark.daly@helsinki.fi (MJD); mrivas@stanford.edu (MAR)

**Data Availability Statement:** The genome-wide summary statistics and the results of gene-based burden and dispersion tests are available at NIH's instance of figshare (https://doi.org/10.35092/yhjc.

## Abstract

Protein-altering variants that are protective against human disease provide *in vivo* validation of therapeutic targets. Here we use genotyping data from UK Biobank (n = 337,151 unrelated White British individuals) and FinnGen (n = 176,899) to conduct a search for protein-altering variants conferring lower intraocular pressure (IOP) and protection against glaucoma. Through rare protein-altering variant association analysis, we find a missense variant in *ANGPTL7* in UK Biobank (rs28991009, p.Gln175His, MAF = 0.8%, genotyped in 82,253 individuals with measured IOP and an independent set of 4,238 glaucoma patients and 250,660 controls) that significantly lowers IOP (β = -0.53 and -0.67 mmHg for heterozygotes, -3.40 and -2.37 mmHg for homozygotes, $P$ = 5.96 x $10^{-9}$ and 1.07 x $10^{-13}$ for corneal compensated and Goldman-correlated IOP, respectively) and is associated with 34% reduced risk of glaucoma ($P$ = 0.0062). In FinnGen, we identify an *ANGPTL7* missense variant at a greater than 50-fold increased frequency in Finland compared with other populations (rs147660927, p.Arg220Cys, MAF Finland = 4.3%), which was genotyped in 6,537 glaucoma patients and 170,362 controls and is associated with a 29% lower glaucoma risk ($P$ = 1.9 x $10^{-12}$ for all glaucoma types and also protection against its subtypes including exfoliation, primary open-angle, and primary angle-closure). We further find three rarer variants in UK Biobank, including a protein-truncating variant, which confer a strong composite lowering of IOP ($P$ = 0.0012 and 0.24 for Goldman-correlated and corneal compensated IOP, respectively), suggesting the protective mechanism likely resides in the loss of interaction or function. Our results support inhibition or down-regulation of ANGPTL7 as a therapeutic strategy for glaucoma.

11368022 and http://doi.org/10.35092/yhjc.
11369166, respectively)[41][47]. Analysis scripts
and notebooks are available on GitHub (https://
github.com/rivas-lab/ANGPTL7/).

**Funding:** Y.T. is supported by a Funai Overseas
Scholarship from the Funai Foundation for
Information Technology and the Stanford
University School of Medicine. FinnGen is
supported by Abbvie, Astra Zeneca, Biogen,
Celgene, Genentech, GSK, Merck, Pfizer, and
Sanofi. M.A.R. is partially supported by Stanford
University and a National Institute of Health center
for Multi- and Trans-ethnic Mapping of Mendelian
and Complex Diseases grant (5U01 HG009080)
and partially supported by the National Human
Genome Research Institute (NHGRI) of the
National Institutes of Health (NIH) under award
R01HG010140. The content is solely the
responsibility of the authors and does not
necessarily represent the official views of the
National Institutes of Health. The funders had no
role in study design, data collection and analysis,
decision to publish, or preparation of the
manuscript.

**Competing interests:** I have read the journal's
policy and the authors of this manuscript have the
following competing interests: Robert Graham is
an employee of MazeTx. The funders had no role in
study design, data collection and analysis, decision
to publish, or preparation of the manuscript.

## Author summary

Glaucoma is a common eye disease that damages the optic nerve. Using intraocular pressure, which is a known modifiable risk factor and predictive measure for glaucoma, genome-wide association studies have identified dozens of genetic variants likely affecting disease risk. However, the identification of potential therapeutic targets from those discoveries has been challenging because the functional consequences and the causal variants of the suggested common variant associations are typically unclear. Here, we present a strategy to scan for rare protein-altering variants, which provides direct insights into the functional consequence and the therapeutic effects, using more than 514,000 individuals with European ancestries in two population cohorts in the UK and Finland. We discover an allelic series of multiple rare *ANGPTL7* missense and nonsense variants in UK Biobank that lower intraocular pressure and reduces the risk of glaucoma. We further identify an *ANGPTL7* missense variant in FinnGen cohort with more than 50-fold enrichment in the Finnish population that provides protection against glaucoma and its subtypes. Our results highlight the benefits of multi-cohort analysis for the discovery of rare protein-altering variants in common diseases and indicate *ANGPTL7* as a therapeutic target for glaucoma.

## Introduction

Intraocular pressure (IOP) is a modifiable risk factor and predictive measure for glaucoma[1–4] (S1 Fig). Genome-wide association studies (GWAS) have commonly used this endophenotype that exhibits high genetic correlation (rg = 0.71) to glaucoma, as an approach to prioritize genetic variants likely to contribute to disease risk[5]. More than 68 independent loci have been implicated with intraocular pressure by meeting the GWAS significance threshold of association ($P < 5x10^{-8}$) [5–8], and a subset have reached genome-wide significance for glaucoma. For these discoveries, like most GWAS results, it has proven challenging to infer the functional consequences of common variant associations beyond cases where protein-altering variants have been directly implicated. Protein-altering variants, generally the strongest-acting genetic variants in medical genetics, include missense substitutions and protein-truncating variants, and understanding their functional consequences provides insight into the therapeutic effects of inhibiting or down-regulating the gene in which they reside[9]. Thus, identifying protein-altering variants that confer protection from disease holds particular promise for identifying therapeutic targets.

Here we leverage two population cohorts that provide complementarity for glaucoma gene discovery (Fig 1). First, UK Biobank has obtained IOP measurements in approximately 128,000 individuals in addition to case-control status for glaucoma from hospital in-patient and verbal questionnaire data in over 500,000 individuals[10–12]. Second, FinnGen has directly genotyped and aggregated disease outcomes in over 176,000 individuals from Finland, an isolated population with recent bottlenecks that offers an unprecedented advantage for studying rare variants in complex diseases[13]. With clinic-based recruitment focused on several areas including ophthalmology, and with 31.1% of the collection above age 70, FinnGen is particularly well-powered for aging-associated endpoints. We, therefore, conduct targeted association analysis with IOP measurements in UK Biobank (N = 82,253, S1 Table) to identify rare protein-altering variants that reduce IOP, and test whether those variants or others in the same genes, also confer protection to glaucoma in FinnGen (6,537 cases and 170,362 controls) and

## Analysis 1: Rare protein-altering variants genome-wide association scan

Intraocular pressure analysis (n = 82,253 individuals)
41,590 genotyped variants; 14,368 genes

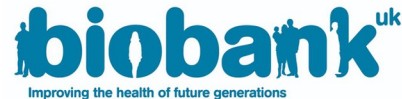

**$P < 10^{-6}$**
*ANGPTL7* p.Gln175His (rs28991009)
1-11253684-G-T
*P* = corneal compensated | Goldman-correlated
Combined Left and Right eye *P* = **$5.96 \times 10^{-9}$** | **$1.07 \times 10^{-13}$**

**$P < 2.5 \times 10^{-6}$**
*ANGPTL7* (burden test, dispersion test) ALL; without p.Gln175His
*P* = corneal compensated | Goldman-correlated
Combined Left and Right eye *P* = (**$1.88 \times 10^{-7}$**, **$1.43 \times 10^{-8}$**); (**0.11, 0.24**) | (**$1.44 \times 10^{-14}$**, **$2.89 \times 10^{-15}$**); (**$3.70 \times 10^{-4}$**, **$1.20 \times 10^{-3}$**)

**Test hypothesis:**
***ANGPTL7* in**
**Glaucoma**

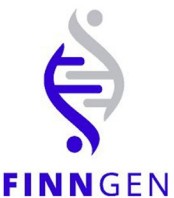

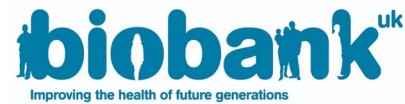

### Analysis 2: Glaucoma analysis
6,537 cases, 170,362 controls

*ANGPTL7* p.Arg220Cys (rs147660927)
1-11253817-C-T
MAF: 4.3% in Finland, 0.1% in UK

| Glaucoma type | Cases | *P* | OR |
|---|---|---|---|
| **ALL** | 6,537 | **$1.9 \times 10^{-12}$** | 0.70 |
| **Primary open-angle** | 3,375 | **$1.3 \times 10^{-8}$** | 0.68 |
| **Primary angle-clos.** | 466 | **0.0016** | 0.59 |
| **Exfoliation** | 1,185 | **$6.7 \times 10^{-5}$** | 0.64 |
| **Normotensive** | 653 | **0.07** | 0.78 |

### Analysis 3: Glaucoma analysis in individuals without IOP measurements (not in Analysis 1)

4,238 cases, 250,660 controls

p.Gln175His (rs28991009), 1-11253684-G-T
*P* = **$6.2 \times 10^{-3}$**, OR = 0.67

*ANGPTL7* (burden test, dispersion test)
ALL; without p. Gln175His
*P* = (**0.0129**, **$8.7 \times 10^{-3}$**) ; (**0.72, 0.72**)

**Fig 1. The overview of the study based on 514,050 individuals in the UK Biobank and FinnGen cohorts.** We identified the association between *ANGPTL7* and intraocular pressure (IOP) phenotypes using the genome-wide association analysis for rare (0.01% < MAF < 1%) protein-altering variants outside of MHC region in UK Biobank and applied burden and dispersion test (Analysis 1). In FinnGen, we discovered a Finnish enriched allele, p.Arg220Cys, in *ANGPTL7* and performed association and subtype analysis of glaucoma (Analysis 2). In the UK Biobank, we replicated the associations between *ANGPTL7* and glaucoma using the individuals that are not included in Analysis 1 (Analysis 3). OR corresponds to the odds ratio.

UK Biobank (4,238 cases and 250,660 controls not included in the initial IOP association analysis). The multi-cohort allelic series analysis of protein-altering variants in *ANGPTL7* in a total of 10,806 glaucoma patients and over 400,000 controls identifies a significant lowering effect on IOP and protective association with glaucoma. By analyzing putative loss-of-function variants, we find concordant effect directions with the missense substitutions, suggesting that the protective mechanism may reside in the loss of gene function.

## Results

We conducted protein-altering variant association analysis with IOP, as measured via corneal-compensated and Goldmann-correlated tonometry, in 82,253 unrelated White British individuals in UK Biobank dataset (S2–S4 Figs, Methods) [14]. Across 41,590 rare (0.01% < MAF < 1%) protein-altering variants outside of the MHC region with genotyping array data in UK Biobank, we performed association analysis to scan for variants with IOP-lowering effects. Specifically, we took the median of the left and right eye IOP measurements, applied quantile normalization and used a generalized linear model implemented in PLINK[13] with age, sex, and the first 4 genotype principal components (PCs) as covariates (Methods). We identified one protein-altering variant significantly associated with lower both IOP measurements below the Bonferroni-corrected $P$ value < $1.0\mathrm{x}10^{-6}$, a missense substitution (p. Gln175His) in *ANGPTL7* ($P = 5.96\mathrm{x}10^{-9}$ and $1.07\mathrm{x}10^{-13}$, $\beta$ = -0.20 and -0.16 SD 95% CI: [-0.21, -0.10] and [-0.25, -0.15], -0.53 and -0.67 mmHg for heterozygotes, -3.40 and -2.37 mmHg for homozygotes for corneal compensated and Goldman-correlated IOP, respectively, Fig 1, Tables 1 and 2).

Based on this finding, we assessed whether any additional rare variant associations in *ANGPTL7* were present. We found three additional independent rare protein-altering variants in *ANGPTL7* (MAF < 0.25%) including a premature stop-gain allele (p.Arg177Ter, MAF = 0.041%, Table 1). Collectively, these three variants showed a marginally significant association with lower one of the IOP measurements ($P$ = 0.24 and 0.0012 for corneal compensated and Goldman-correlated IOP, respectively, Fig 1), with the protein-altering allele p.Arg140His also showing a marginally significant effect on its own ($P = 1.3\mathrm{x}10^{-3}$ and $1.9\mathrm{x}10^{-3}$ for corneal compensated and Goldman-correlated IOP, respectively, Table 1). Genotyping intensity plots and the concordance of genotype calls from the array and whole-exome sequencing data were manually inspected to ensure high quality and consistent genotyping (S5 Fig, S2 Table, Methods) and alleles were confirmed to be independent (pairwise $r^2 < 10^{-4}$ for each, S3 Table, Methods). The burden and dispersion test showed significant p-values for *ANGPTL7* ($P = 1.88\mathrm{x}10^{-7}$ [burden], $1.43\mathrm{x}10^{-8}$ [dispersion]; and $1.44\mathrm{x}10^{-14}$ [burden], $2.89\mathrm{x}10^{-15}$ [dispersion] for corneal compensated and Goldman-correlated IOP, respectively, Fig 1). Collectively, those four alleles explain 0.03% and 0.07% of phenotypic variation for corneal-compensated and Goldmann-correlated tonometry IOP measures (S4 Table). These signals were consistently observed in corneal-compensated and Goldmann-

**Table 1. *ANGPTL7* IOP protein-altering variant association in UK Biobank.** The association statistics for corneal compensated IOP and Goldman-correlated IOP are shown. Variant includes chromosome, position, reference, and alternate allele (hg19). rsID—the rs identifier of the genetic variant. HGVSp—the HGVS amino acid nomenclature. MAF—the minor allele frequency in UK Biobank white British population. Beta—estimated regression coefficient with 95% confidence intervals. *P*—p-value of association.

| Variant (rsID) | HGVSp | MAF (UKB) | corneal compensated IOP (INI2005254) | | Goldman-correlated IOP (INI2005255) | |
| --- | --- | --- | --- | --- | --- | --- |
| | | | Beta SD [95% CI] | *P* | Beta SD [95% CI] | *P* |
| 1:11252357:A:G (rs200058074) | p.Gln136Arg | .054% | 0.012 [-0.20, 0.23] | $9.1\mathrm{x}10^{-1}$ | -0.030 [-0.25, 0.19] | $7.8\mathrm{x}10^{-1}$ |
| 1:11252369:G:A (rs28991002) | p.Arg140His | .25% | -0.071 [-0.17, 0.022] | $1.3\mathrm{x}10^{-3}$ | -0.15 [-0.24, -0.055] | $1.9\mathrm{x}10^{-3}$ |
| 1:11253684:G:T (rs28991009) | p.Gln175His | .81% | -0.16 [-0.21, -0.10] | $6.0\mathrm{x}10^{-9}$ | -0.20 [-0.25, -0.15] | $1.1\mathrm{x}10^{-13}$ |
| 1:11253688:C:T (rs143435072) | p.Arg177Ter | .041% | -0.13 [-0.37, 0.12] | $3.0\mathrm{x}10^{-1}$ | -0.29 [-0.53, -0.038] | $2.4\mathrm{x}10^{-2}$ |

**Table 2. *ANGPTL7* allelic series association summary in UK Biobank and FinnGen.** Variant: the rs identifier (rsID), the amino acid nomenclature (HGVSp), and genomic coordinate (CHR for chromosome and POS for the position in hg19), as well as reference (REF) and alternate (ALT) alleles, are shown. Dosage—genotype of individuals for protein and nucleotide sequences. Carrier frequency—carrier frequency in UK Biobank (UK) and FinnGen (Finland) for the respective genotype dosage. N with IOP—number of individuals in UK Biobank with intraocular pressure measurements corresponding to the genotype dosage. Effect size estimates—reported effect size estimates. IOP (mmHg) [95% CI]—unstandardized estimates of effect size on corneal-compensated and Goldmann-correlated intraocular pressure measurements (NB: standardized estimated effect sizes may have lower p-values due to normalization procedure). OR for glaucoma—estimate odds ratio on glaucoma risk for the respective genotype dosage. NS non-significant (p > 0.1). Effect sizes always reported with respect to alternate allele dosage.

| Variant | Dosage | Carrier frequency | | N with IOP in UK | Effect size estimates [95% CI] | | |
| --- | --- | --- | --- | --- | --- | --- | --- |
| rsID HGVSp CHR:POS:REF:ALT | Protein nucleotide | UK | Finland | | IOP (mmHg) | | OR for glaucoma |
| | | | | | corneal-compensated (INI2005254) | Goldmann-correlated (INI2005255) | |
| rs200058074 p.Gln136Arg 1:11252357:A:G | Gln/Arg A/G | 0.11% | NA | 80 | -0.051 [-0.89, 0.79] NS | -0.19 [-0.99, 0.61] NS | NS |
| rs28991002 p.Arg140His 1:11252369:G:A | Arg/His G/A | 0.51% | 0.35% | 427 | -0.23 [-0.59, 0.14] NS | -0.51 [-0.86, -0.17] ** | NS |
| rs28991009 p.Gln175His 1:11253684:G:T | Gln/His G/T | 1.43% | 0.24% | 1355 | -0.53 [-0.73, -0.32] *** | -0.67 [-0.87, -0.47] *** | 0.64 [0.48, 0.87] ** |
| | His/His T/T | 0.01% | NA | 5 | -3.40 [-6.8, -0.042* | -2.37 [-5.6, 0.84] NS | NS |
| rs143435072 p.Arg177Ter 1:11253688:C:T | Arg/Ter C/T | 0.07% | NA | 62 | -0.46 [-1.4, 0.5] NS | -0.95 [-1.9, -0.034] * | NS |
| rs147660927 p.Arg220Cys 1:11253817:C:T | Arg/Cys C/T | NA | 7.82% | NA | NA | NA | 0.67 *** |
| | Cys/Cys T/T | NA | 0.21% | NA | NA | NA | 0.31 * |

Significant codes: 0 '***' 0.001 '**' 0.01 '*' 0.05 '.' 0.1 'NS' 1

correlated tonometry IOP measures for both left and right eyes (S5 Table), which is expected as the genetic correlation among those range from 0.75 to 1.0 (S6 Table).

In Khawaja et al. the authors identified a significant association in the same genomic region (chr1p36.22), which they label as the *UBIAD1* region [6]. The associated region, defined by recombination rates, encompasses a relatively large area around SNP rs143038218 and includes the *ANGPTL7* gene. We asked whether this signal could be explained by our allelic series of protein-altering variants in *ANGPTL7*. Linkage disequilibrium analysis shows that p.Gln175His has an $r^2$ = 0.83 suggesting that the *ANGPTL7* protein-altering variant was indeed responsible for the association signal observed in Khawaja et al, whereas p.Arg177Ter, p.Arg140His, and p.Gln136Arg all have $r^2$ of approximately 0 and are under linkage-equilibrium as minor alleles are observed in different haplotypes and contribute independently to the association we observe in *ANGPTL7* against IOP.

We next asked whether any of these putative IOP-lowering genetic variants showed effects consistent with reducing glaucoma risk. We focused on unrelated White British individuals that do not have IOP measures (4,238 cases and 250,660 controls, S6 Fig). For p.Gln175His in *ANGPTL7*, using logistic regression analysis with age, sex, and principal components (PC1-PC4) as covariates, we estimated that the variant lowers glaucoma risk by 34% (*P* = 0.00543; OR = 0.66 [95% CI: 0.366–0.954], Table 2). The three additional protein-altering variants did not significantly confer protection against glaucoma (burden test *P* = 0.77). This is consistent with power calculations, using Genetic Power Calculator [15], where our power to detect association for rare

variants with a composite allele frequency of 0.345% and a binary trait in 4,238 cases and 250,660 controls at alpha = 0.05, and 0.001, i.e. $P < .05$ and $.001$, with OR of 0.7 (30% reduction in risk) is equal to 62% and 15.2%, respectively.

We then sought evidence from the FinnGen dataset that either the same or novel Finnish-enriched protein-altering variants would confirm the association of *ANGPTL7* with protection from glaucoma. This additional *ANGPTL7* association data, obtained in 6,537 glaucoma patients and 170,362 controls, provided strong support that protein-altering variants in *ANGPTL7* protect against glaucoma (case definitions described in S7 Table). Specifically, we found that the p.Gln175His substitution has nominal evidence of association ($P = 0.009$, OR = 0.47) despite the variant only being present at a minor allele frequency of 0.1% in this Finnish cohort (8-fold depleted compared to UK Biobank). The remaining protein-altering variants in *ANGPTL7* tested in UK Biobank were not found in the FinnGen dataset. Confirmation of an *ANGPTL7* effect on glaucoma risk was seen in data from an independent Finnish-specific protein-altering missense substitution, p.Arg220Cys, which was strongly associated with protection from glaucoma ($P = 2.0 \times 10^{-12}$, OR = 0.71 [95% CI: 0.64–0.78], S7 Fig). Of note, this observation is advantaged by the property that p.Arg220Cys is found at a greater than 50-fold increased frequency in Finland compared with other populations [16], reinforcing the value of isolated, bottlenecked populations in which the allele frequency spectrum is intensely concentrated on the minority of variants passing through the bottleneck.

While registry-based diagnoses in FinnGen do not yet contain detailed ophthalmologic records, a subset of 3,375 glaucoma patients had been recorded in special health care as having primary open-angle glaucoma (POAG). In this sub-group, a stronger effect was observed ($P = 1.3 \times 10^{-8}$, OR = 0.68) versus those glaucoma cases without a definitive record of POAG (OR = 0.77 [95% CI: 0.67–0.88]), reminiscent of the stronger risk effects seen at the myocilin (*MYOC*) gene and other established genes in the POAG subgroup [17]. Furthermore, we find protective association to exfoliation glaucoma ($P = 6.7 \times 10^{-5}$, OR = 0.64), primary angle-closure glaucoma ($P = 0.0016$, OR = 0.59), and normotensive glaucoma ($P = 0.07$, OR = 0.78, case n = 653, Fig 1).

Given the Finnish enrichment of the known strong glaucoma risk allele, p.Gln368Ter, in *MYOC* (MAF in Finland = 0.3%, MAF in Non-Finnish European = 0.16%, reference sequence: NM_00026), we next asked whether carriers have risk reduced if they carry *ANGPTL7* p.Arg220Cys. In FinnGen, we estimate that 7.0% of carriers for *MYOC* p.Gln368Ter variant is POAG cases in comparison to 2% for non-carriers. In the presence of *ANGPTL7* p.Arg220Cys, only 1.3% of individuals are POAG cases, and only 2 of 86 (2.3%) who carry both *MYOC* risk and *ANGPTL7* protective variants were POAG cases (S8 Table). This suggests *ANGPTL7* protection extends to the *MYOC* risk group but the small counts preclude any definitive statement regarding interaction ($P = 0.318$, for interaction term in a logistic regression model)—given the limited number of double-carriers, larger case-control series are needed to refine our understanding as to whether *ANGPTL7* p.Arg220Cys variant modifies the glaucoma risk conferred by p.Gln368Ter in *MYOC*.

Access to genotype data in over 330,000 individuals from the UK and 176,000 from the Finnish group enabled us to identify rare protein-altering homozygotes. In UK Biobank, we found 28 individuals homozygous for the p.Gln175His allele, consistent with Hardy-Weinberg expectation (n = 22.6), where we estimated a -3.40 and -2.37 mmHg drop for corneal compensated and Goldman-correlated IOP, respectively, compared to the mean IOP levels. Furthermore, the oldest reached the age of 80 and one of the 28 died (age 65). In FinnGen we found 343 individuals homozygous for the p.Arg220Cys allele, the oldest reached the age of 98, with no depletion of homozygotes compared with Hardy-Weinberg equilibrium expectation. There was no significant association of the homozygous genotype with a decreased lifespan. We did

not observe homozygous p.Arg220Cys association across disease endpoints in FinnGen (Bonferroni-corrected p > 0.05, S1 Data). Together this indicates that having two copies with p. Gln175His or p.Arg220Cys in *ANGPTL7* is compatible with a normal lifespan. To assess the potential impacts of those protein-truncating variants on reproductive fitness, we assessed the association of p.Gln175His with the number of live births and the number of children fathered and found no significant association (*P* > 0.05/4, S9 Table). Through phenome-wide association analysis (PheWAS), we did not find any significant association for non-eye phenotypes (P>1.0 x $10^{-5}$ for both in UK Biobank and FinnGen, S10 Table, S2 and S3 Data). Hence, we did not find any severe medical consequences that would be of obvious concern in developing a therapeutic to mimic the effect of these alleles.

*ANGPTL7*, a five-exon protein-coding gene, encodes the Angiopoietin-related protein 7, which is expressed in several human tissues including the trabecular meshwork, cornea, and retina [18–20]. We examined proteomics expression data in normal tissues and cell lines from ProteomicsDB and MOPED[21,22] and found vitreous humor tissue-specific expression of ANGPTL7 (log10 ppm = 1, S8 Fig).

## Discussion

This study establishes strong genetic evidence for the involvement of *ANGPTL7* in glaucoma risk in which a powerful allelic series, including multiple low-frequency missense substitutions and a single premature stop-gain substitution, is conclusively associated with reduced disease risk and endophenotype-lowering effects. Our results highlight the benefit of rare protein-altering variant analysis using multiple large cohorts, especially when the population history of the participating cohort experienced a bottleneck, which enables enrichment of rare alleles as we report with the *ANGPTL7* p.Arg220Cyc allele [13]. In Finland, the most common glaucoma subtypes are POAG and the secondary exfoliation glaucoma. The main difference in glaucoma prevalence is that in Finland the exfoliation glaucoma is much more prevalent (31%) than in the UK [23]. The prevalence of POAG is similar in Finland than in other European populations. The prevalence is heavily affected by age. In one Finnish cohort study, among individuals aged 70 years or older, the prevalence of POAG was approximately 7% [24]. Relative similar prevalence for POAG is reported in European populations [25]. Many patients with POAG are undiagnosed so the prevalence is affected by sampling methods (i.e. cohort or diagnosis reported). The population cohorts from founder populations enable future recall studies focusing on individuals homozygous for the allele, which can eventually improve our understanding of the mechanism by which ANGPTL7 disruption leads to protection to glaucoma risk and lowering of IOP. The discovery of two independent protein-altering alleles with directionally consistent effects from the two analyzed populations increases our confidence in the gene's causal link to glaucoma.

Recent studies have associated ANGPTL proteins with cardiometabolic phenotypes [26–30]. Although it has been proposed that ANGPTL7 levels are increased in obesity (and reduced after physical exercise), we do not observe any evidence of genetic association in either UK Biobank or FinnGen to support this hypothesis [31].

*ANGPTL7* overexpression in primary human trabecular meshwork cells was found to alter the expression of relevant trabecular meshwork proteins of the extracellular matrix, including fibronectin, collagens type I, IV, and V, myocilin, versican, and MMP1, and ANGPTL7 protein was increased as the disease progressed in POAG beagle dogs [18]. The tissue-specific protein expression data suggest that further work in dissecting the role of *ANGPL7* in all possible cell types in the eye is warranted.

When combined with the previously-reported associations with IOP and glaucoma, our results provide compelling genetic evidence of the role of ANGPTL7 in glaucoma and its sub-types including exfoliation, primary open-angle, and primary angle-closure, which may come in contrast to prior findings with lack of overlap between POAG risk and IOP loci [32]. In the context of the other established variants in glaucoma, including the protein-truncating variants in *MYOC*, p.Gln175His and the 57-fold Finnish-enriched p.Arg220Cys variants in *ANGPTL7* exert a comparable protective effect. While our genetic discovery provides compelling evidence of involvement of *ANGPTL7* in glaucoma, several important questions remain to be answered before its eventual clinical translation. First, we were not able to assess whether the missense variants are complete loss-of-function, partial loss-of-function variants, dominant negative, or gain of function given the data we have at hand. Although we do have a predicted protein-truncating variant, p.Arg177Ter, with nominal evidence of association to IOP and an estimated effect consistent with the missense substitutions, it is challenging to draw conclusions about its functional consequence from *in silico* predictions, as we have reported in earlier studies assessing when PTVs trigger degradation pathways like nonsense-mediated decay [33]. Second, it is unclear in which cell types these variants are acting on to confer the protective and IOP lowering effects. We anticipate that ANGPTL7 may be acting in the trabecular mesh-work given its high expression in both adult and fetal trabecular meshwork ($>$ 3000 FPKM) [34], we see high expression in both adult and fetal cornea ($>$200 FPKM), which introduces some challenges as how we interpret its functional role, and we hypothesize that given its high expression in cornea it may be one reason why we see stronger evidence of association in IOP Goldman correlated measures compared to corneal compensated IOP. Additionally, future studies should assess whether *ANGPTL7* variants modify the progression of glaucoma, for example, whether *ANGPTL7* carriers are less likely to go from glaucoma diagnosis to potential surgery. Although we are aggregating these data, we are thus far unable to draw definitive conclusions.

Because of the strong protective effect associated with the *ANGPTL7* protein-altering variants (S9 Fig), further studies of ANGPTL7 inhibition and the specific action of these variant proteins should be useful in understanding the mechanism by which glaucoma protection occurs and whether this reveals a promising therapeutic opportunity similar to that which has been realized from the examples of *PCSK9*, *APOC3* and cardiovascular disease [35–37]. Given the rapidly evolving field of gene editing and siRNA, we can only speculate that if the effect is truly loss-of-function and that gene inhibition is an appropriate strategy then these therapeutic modalities will be especially relevant. Therapeutic delivery is also a complicated challenge. Although injection to the eye is currently commonplace in practice, it is unclear whether different therapeutic modalities, e.g. antibody, siRNA, CRISPR, base-editing would be appropriate, and whether the duration of the treatment would be sufficiently durable to be effective to prevent extremely frequent injections or competitive against current therapeutic modalities. New drug delivery technologies are of interest and it is clear that a durable and efficient mode of delivery that mimics the protective effect of these mutations is an attractive strategy. Our genetic data from *ANGPTL7* homozygotes with up to a 69% risk reduction for all glaucoma and 80% risk reduction for primary open-angle glaucoma suggest that this is likely to be a safe and effective strategy for therapeutic intervention.

## Methods

### Compliance with ethical regulations and informed consent

This research has been conducted using the UK Biobank Resource under Application Number 24983, "Generating effective therapeutic hypotheses from genomic and hospital linkage data"

(criteriahttp://www.ukbiobank.ac.uk/wp-content/uploads/2017/06/24983-Dr-Manuel-Rivas. pdf). Based on the information provided in Protocol 44532 the Stanford IRB has determined that the research does not involve human subjects as defined in 45 CFR 46.102(f) or 21 CFR 50.3(g). All participants of UK Biobank provided written informed consent (more information is available at https://www.ukbiobank.ac.uk/2018/02/gdpr/). For the Finnish Institute of Health and Welfare (THL) driven FinnGen preparatory project (here called FinnGen), all patients and control subjects had provided informed consent for biobank research, based on the Finnish Biobank Act. Alternatively, older cohorts were based on study specific consents and later transferred to the THL Biobank after approval by Valvira, the National Supervisory Authority for Welfare and Health. Recruitment protocols followed the biobank protocols approved by Valvira. The Ethical Review Board of the Hospital District of Helsinki and Uusimaa approved the FinnGen study protocol Nr HUS/990/2017. The FinnGen preparatory project is approved by THL, approval numbers THL/2031/6.02.00/2017, amendments THL/341/ 6.02.00/2018, THL/2222/6.02.00/2018 and THL/283/6.02.00/2019. All DNA samples and data in this study were pseudonymized.

## Genome-wide association analysis in UK Biobank

**Population stratification in UK Biobank.** We used genotype data from the UK Biobank dataset release version 2 and the hg19 human genome reference for all analyses in the study. To minimize the variabilities due to population structure in our dataset, we restricted our analyses to include 337,151 White British individuals (S2 Fig) based on the following five criteria [11,38] reported by the UK Biobank in the file "ukb_sqc_v2.txt":

1. self- reported white British ancestry ("in_white_British_ancestry_subset" column)

2. used to compute principal components ("used_in_pca_calculation" column)

3. not marked as outliers for heterozygosity and missing rates ("het_missing_outliers" column)

4. do not show putative sex chromosome aneuploidy ("putative_sex_chromo- some_aneuploidy" column)

5. have at most 10 putative third-degree relatives ("excess_relatives" column).

Of note, we included the entire age range of the UK Biobank cohort for our analysis to maximize the power of association analysis.

**Intraocular pressure phenotype definitions in UK Biobank.** We focused on Goldmanncorrelated and corneal-compensated IOP measurements of left and right eyes from UK Biobank (Field IDs: 5254, 5255, 5262, and 5263, S1 Table). For each field, there were up to two measurements, which corresponds to the initial assessment visit (2006–2010) and the first repeat assessment visit (2012–13). We additionally defined the median Goldmann-correlated and corneal-compensated IOP phenotypes by taking the median of up to 4 measurements for each (Global Biobank Engine phenotype IDs: INI2005254 and INI2005255, S1 Table), combining the left and right eye measurements.

**Rare protein-altering variant genome-wide association scan for IOP.** For the white British individuals (n = 337,151) in UK Biobank [11], we applied genome-wide association analysis for directly genotyped variants and phenotypes with inverse-normal transformation (‑‑pheno-quantile-normalize option) using generalized linear regression model implemented in PLINK v2.00aLM (12 Nov. 2019) with age, sex, types of genotyping array, and the first 4 genotype principal components, where array is an indicator variable that indicates whether the

individual was genotyped using the UK BiLEVE array or the UK Biobank array, as described elsewhere [38,39]. The inverse-normal transformation (--pheno-quantile-normalize option in PLINK2) is a non-parametric phenotype normalization procedure and it forces the phenotype to a standard normal distribution, preserving just the quantiles. For example, if the original phenotype values are 9, 4, 9, and 7 in that order, the quantiles are 0.75, 0.125, 0.75, 0.375, and the transformed phenotype values are the inverse-normal-cdf of each of the quantile value (https://www.cog-genomics.org/plink/2.0/data#quantile_normalize). The genome-wide association summary statistics are available at NIH's instance of figshare.

**Glaucoma association analysis in individuals without IOP measurements.** To assess the potential effects of identified putative IOP-lowering genetic variants on glaucoma risk, we applied the genome-wide association analysis for glaucoma (Global Biobank Engine phenotype ID: HC276) focusing on 254,898 individuals (4,238 cases and 250,660 controls) in UK Biobank who do not have any of the IOP measurements (Fig 1). The glaucoma phenotype was previously defined as a part of "high confidence" disease outcome phenotypes by combining disease diagnoses (UK Biobank Field ID 41202, 41204, 40001, and 40002) from the UK National Health Service Hospital Episode Statistics (ICD10 codes: H40.[0–6,8,9], H42.8, and Q15.0) with self-reported non-cancer diagnosis questionnaire (UK Biobank Field ID 20002), as summarized as an UpSet plot in S6 Fig [11,12,40].

We used logistic regression with the firth-fallback option using a generalized linear regression model implemented in PLINK v2.00aLM (12 Nov. 2019) with age, sex, types of genotyping array, and the first 4 genotype principal components. The genome-wide association summary statistics are available at NIH's instance of figshare[41].

**Targeted regression analysis of identified rare variants in *ANGPTL7*.** To assess the impacts of identified rare variants in *ANGPTL7* on unnormalized IOP, we performed linear regression for IOP. Specifically, we used the following formula and called the linear model implemented in R.

lm (IOP ~ age + as.factor(sex) + as.factor(Array) + PC1 + PC2 + PC3 + PC4, as.factor (SNV), family = binomial(link = "logit"))

## Genotyping quality control in UK Biobank

**Manual inspection of intensity plots.** For the identified rare (0.01% < MAF < 1%) protein-altering variants in *ANGPTL7* (reference sequence: NM_021146), we generated and inspected intensity plots with McCarthy Group's ScatterShot using "UKB—All Participants" module [42].

**Variant-calling consistency analysis.** For individuals with whole-exome sequencing data (n = 49,960), we extracted the genotype calls of coding variants in *ANGPTL7* using PLINK v2.00aLM (2 April 2019) and compared the consistency between the array-genotyped and whole-exome sequencing dataset [39,43].

## Burden and dispersion tests of rare protein-altering variants

To assess associations with rare protein-altering variants, we performed a burden and dispersion test implemented in multiple rare variants and phenotypes (MRP) package with farebrother option (https://github.com/rivas-lab/ANGPTL7/tree/master/gene_based_test) [44,45]. The approach implemented in the MRP package is a generalization of the gene-based test for a single phenotype described in the Supplementary Material of Band et al. [46] *Region-based test* and subsection labeled *calculating p-values*. We used the GWAS summary statistics of rare (0.01% < MAF < 1%) protein-altering variants characterized form the procedure above as the

input data and performed the genome-wide burden and dispersion tests. The results of the burden and dispersion analysis are publicly available at NIH's instance of figshare[47].

## Independence analysis of alleles

**Pairwise $r^2$ computation within British individuals in UK Biobank.** We computed pairwise $r^2$ for the identified rare protein-altering variants in *ANGPTL7* within British individuals in UK Biobank using PLINK v1.90b6.7 64-bit (2 Dec 2018) with--ld <variant_ID_1> <variant_ID_1> hwe-midp subcommand [39].

**Number of individuals with the combination of genotypes in UK Biobank.** Using the extracted genotype calls from for the identified rare protein-altering variants in *ANGPTL7* (see Variant-calling consistency analysis section), we counted the number of British individuals by the combination of genotypes. We computed the expected number of individuals under Hardy-Weinberg equilibrium model and the independence assumption:

- The expected frequencies of REF/REF, REF/ALT, and ALT/ALT carriers are $(1-AF)^2$, 2 x AF(1-AF), and $AF^2$, respectively.

- The expected genotyping rate is independently estimated by the observed genotyping rate for each variant.

- The expected frequency of the combination of genotypes is computed under the independent assumption among alleles

## Local heritability analysis

To estimate the proportion of phenotypic variation explained by the rare protein-altering variants in *ANGPTL7*, we used Haseman-Elston (HE) regression using the cross product of the phenotypes for pairwise individuals implemented in genome-wide complex trait analysis (GCTA) version 1.92.4beta2 [48,49]. We computed the genetic relationship matrix (GRM) using the 4 rare protein-altering variants in *ANGPTL7* and used it for the HE regression analysis [50].

## Genetic correlation analysis

To estimate the genetic correlation, we used bivariate-HEreg using the cross product of the phenotypes for pairwise individuals implemented in GCTA version 1.92.4beta2. We computed GRM based on non-rare (MAF > 1%) variants on the genotyping array and used it for the bivariate-HEreg analysis.

## PheWAS-analysis in UK Biobank

Using the summary statistics that are previously described and hosted on Global Biobank Engine (GBE) [11,38,51], we performed the phenome-wide association studies for 173 disease outcomes in the UK Biobank (https://github.com/rivas-lab/ANGPTL7/blob/master/notebook/ukbb_phewas/phenotypes_used_for_PheWAS.txt). Briefly, the summary statistics are generated by linear regression (for continuous traits) or logistic regression with the firth-fallback option (for binary outcomes) using the--glm subcommand implemented in PLINK v2.00a with age, sex, and the first 4 genotype PCs as covariates. We summarized the associations with $P < 1$x$10^{-4}$ and SE < 0.5 (S10 Table). The full PheWAS association results are available on Global Biobank Engine [51].

- https://gbe.stanford.edu/RIVAS_HG19/variant/1-11252357-A-G

- https://gbe.stanford.edu/RIVAS_HG19/variant/1-11252369-G-A

- https://gbe.stanford.edu/RIVAS_HG19/variant/1-11253684-G-T

- https://gbe.stanford.edu/RIVAS_HG19/variant/1-11253688-C-T

### PheWAS-analysis in FinnGen

In FinnGen, we performed a phenome-wide association analysis (PheWAS) of the identified variant comprising of 2,264 disease endpoints (S7 Fig, S2 and S3 Data).

**Disease endpoint definition in FinnGen.** The disease endpoints were defined using nationwide registries for deaths, hospital discharges, outpatient specialist appointments, cancer registry, and drug purchases registry, harmonizing over the International Classification of Diseases (ICD) revisions 8, 9, and 10, cancer-specific ICD-O-3, (NOMESCO) procedure codes, Finnish-specific Social Insurance Institute (KELA) drug reimbursement codes and ATC-codes. These registries spanning decades were electronically linked to the cohort baseline data using the unique national personal identification numbers assigned to all Finnish citizens and residents. A full list of FinnGen endpoints is available online for Freeze 4 (https://www.finngen.fi/en/researchers/clinical-endpoints). The endpoints with fewer than 100 cases, near-duplicate endpoints, and developmental "helper" endpoints were excluded from the final PheWas (column "OMIT").

**Outlier removal and PCA in FinnGen R4.** FinnGen data was combined with 1000 genomes data and samples of non-Finnish ancestry (n = 2,880) and duplicates (n = 2035) were removed. King software [52] was used for relationship inference and approximately (> = 3rd degree related) independent set of 131,863 samples and 36,944 good quality (variant filters: remove chromosome X, imputation info< = 0.95, genotype imputed posterior probability<0.95, missingess>0.01) LD-pruned (r2<0.1) common (MAF > = 0.05) variants were used for computing PCA with Plink 1.9 [39]. The remaining 46,916 samples were then projected onto those PCs. Further 1,880 samples were removed due to missing phenotype data or mismatching sex and 176,899 samples were used in the analysis.

**Association analysis.** SAIGE mixed-model logistic regression was used for association analysis. Age, sex, 10 PCs and genotyping batch were used as covariates [53]. Each genotyping batch was included as a covariate for an endpoint if there were at least 10 cases and 10 controls in that batch to avoid convergence issues. Variants with minimum allele count < = 5 or imputation info < = 0.6 were excluded from the analysis. We report associations with $P < 1 \times 10^{-4}$ (S7 Fig, S1 and S2 Data).

### Interaction analysis of *ANGPTL7* and *MYOC*

To assess whether there is an interaction between *ANGPTL7* and *MYOC*, we performed a logistic regression analysis using R glm() function with binomial response and logit link function with an interaction term, i.e. *ANGPTL7* x *MYOC*. We found no evidence of interaction effect, $P = 0.318$.

### Association analysis with reproductive fitness

Using the number of live births (UK Biobank Field ID: 2734, Global Biobank Engine phenotype ID: INI2734) and the number of children fathered (UK Biobank Field ID: 2405, Global Biobank Engine phenotype ID: INI2405), we performed association analysis for the four

identified protein-altering variants using R script with age, types of genotyping array, and the first 4 genotype principal components as covariates. The analysis script is available at the GitHub repository (https://github.com/rivas-lab/ANGPTL7/).

### Homozygote analysis

**Homozygote in UK Biobank.** For UK Biobank British individuals, we extracted the genotype calls with PLINK v2.00aLM (2 April 2019) and identified homozygous carrier of p. Gln175His allele [39]. We examined the year of birth (UK Biobank Field ID 34) and age at death (UK Biobank Field ID 40007) [10].

**Homozygote in FinnGen.** For Finnish FinnGen individuals, we extracted the genotypes using bcftools v1.9 and identified homozygote carriers of p.Arg220Cys [54]. To examine Finn-Gen disease endpoints among homozygote variant carriers, we compared the number disease endpoint cases in homozygote individuals to the number of cases of in the FinnGen samples using Fisher's exact test (S1 Data).

### Cascade plot analysis

**Cascade plot of IOP association statistics in UK Biobank.** Using the genome-wide association summary statistics for the median corneal compensated and Goldman-correlated IOP measurements in UK Biobank, we plotted the minor allele frequency and the BETA (SD) for the LD-pruned variants with $P < 5 \times 10^{-8}$. The LD pruning was performed using PLINK 1.9 with "--indep 50 5 2" as described before [11,38].

**Effect size comparison of glaucoma associations with cascade plot.** We collected the previously described association statistics from the following tables in literature.

- Choquet, et al. 2018 [55], Table 2 and Table 3

- Khawaja, et al. 2018 [6], Table 1

- Shiga, et al. 2018 [56], Table 1 and Table 2

- MacGregor, et al. 2018 [5], Supplementary Table S1

- Hysi, et al. 2014 [8], Table 1

We plotted the minor allele frequency and odds ratio for variants with $P < 5 \times 10^{-8}$ for glaucoma.

## Supporting information

**S1 Text. List of FinnGen members.** FinnGen consists of the people listed in the Supplementary text.
(DOCX)

**S1 Fig. Phenotype distributions of intraocular pressure measurements.** Phenotype distributions of the corneal-compensated (A) and Goldman-correlated (B) IOP measurements (the median of left and right eyes) stratified by glaucoma disease status in unrelated White British in UK Biobank displayed as a Tukey's box plot overlapping on a violin plot. In the box plot, the middle bold horizontal line represents the median, the lower and upper hinges show the first and third quartiles, the lower and upper whiskers represent 1.5 * interquartile range from the hinges. The data points beyond whiskers are plotted individually.
(TIF)

**S2 Fig. The identification of unrelated White British individuals in UK Biobank.** The identification of unrelated White British individuals in UK Biobank. The first two genotype principal components (PCs) are shown on the x- and y-axis and the identified unrelated White British individuals (Methods) are shown in red.
(TIF)

**S3 Fig. Genome-wide protein-altering variant association analysis of intraocular pressure phenotypes in UK Biobank.** Genome-wide protein-altering variant association analysis of corneal compensated (A) and Goldman-correlated (B) intraocular pressure in UK Biobank. The rare (0.01% < MAF < 1%) protein-altering variants with P < 0.01 are shown. The red dashed horizontal line represents the genome-wide significance threshold (P = $10^{-6}$). The variants are shown in red (odd autosomes) or blue (even autosomes). The genomic coordinates of the variants are shown on the x-axis and the statistical significance of univariate analysis is shown on the y-axis.
(TIF)

**S4 Fig. The protein-altering variant GWAS QQ plot for intraocular pressure phenotypes.** The protein-altering variant GWAS QQ plot for corneal compensated (A) and Goldman-correlated (B) intraocular pressure. The variants outside of MHC region with 0.01% < MAF < 1% are included in the analysis.
(TIF)

**S5 Fig. The intensity plots for *ANGPTL7* protein-altering variants with 0.01% < MAF < 1%.** The intensity plots for *ANGPTL7* protein-altering variants with 0.01% < MAF < 1%. **(A)** rs200058074 (p.Gln136Arg). **(B)** rs28991002 (p.Arg140His). **(C)** rs28991009 (p.Gln175His). **(D)** rs143435072 (p.Arg177Ter).
(TIF)

**S6 Fig. The breakdown of the data sources used for the definition of glaucoma in UK Biobank.** The breakdown of the data sources used for the definition of glaucoma in UK Biobank. The combination of self-reported glaucoma (coded as "1277" in UKB Data coding ID 6) and ICD-10 codes from hospital inpatient data are used for the glaucoma definition in UK Biobank. The number of individuals in the white British individuals without IOP measurements are shown.
(TIF)

**S7 Fig. Phenome-wide association analysis of p.Arg220Cys in FinnGen.** Phenome-wide association analysis of p.Arg220Cys in FinnGen. -log10(P-value) is displayed on the y-axis. Disease endpoints grouped by disease categories are displayed on the x-axis. Highlighted associations with $P < 1 \times 10^{-4}$ are shown.
(TIF)

**S8 Fig. Protein expression in normal tissues and cell lines from ProteomicsDB and MOPED for ANGPTL7.** Protein expression in normal tissues and cell lines from ProteomicsDB and MOPED for ANGPTL7.
(TIF)

**S9 Fig. The cascade plots for intraocular pressure and glaucoma.** The cascade plot for corneal compensated (A) and Goldman-correlated (B) intraocular pressure association analysis in UK Biobank. The cascade plot for glaucoma (C) from published genome-wide significant GWAS associations (gray) and the variants highlighted in our paper. The minor allele frequency and the BETA (SD) are plotted for the LD-pruned variants with $P < 5 \times 10^{-8}$. The odds

ratios are included for LD pruned published variants with $P < 5x10^{-8}$ for glaucoma.
(TIF)

**S1 Data. The homozygous carrier analysis results for *ANGPTL7* p.Arg220Cys allele in FinnGen.** The homozygous carrier analysis results for *ANGPTL7* p.Arg220Cys allele in Finn-Gen. The phenotype (Phenotype and Phenotype_description), the number of cases and controls and case frequency in homozygous carriers (HOM_case, HOM_cntrl, and HOM_case_%, respectively), the number of cases and controls and case frequency in all individuals (ALL_case, ALL_cntrl, and ALL_case_%, respectively), odds ratio (odds_ratio), and p-value from Fisher's exact test (Fisher_p-value) are shown for disease endpoints.
(XLSX)

**S2 Data. The PheWAS results for *ANGPTL7* p.Arg220Cys allele for eye-related phenotypes in FinnGen.** The PheWAS results for *ANGPTL7* p.Arg220Cys allele for eye-related phenotypes in FinnGen. The phenotype (phenotype_code and phenotype_string), phenotype category (category), beta value (beta), odds ratio (odds_ratio), and p-value (pval) of the association are shown for each of the eye phenotypes.
(XLSX)

**S3 Data. The PheWAS results for *ANGPTL7* p.Arg220Cys allele for non-eye related phenotypes in FinnGen.** The PheWAS results for *ANGPTL7* p.Arg220Cys allele for non-eye related phenotypes in FinnGen. The phenotype (phenotype_code and phenotype_string), phenotype category (category), beta value (beta), odds ratio (odds_ratio), and p-value (pval) of the association are shown for each of the phenotypes that are not labeled as "Diseases of the eye and adnexa" in phenotype category.
(XLSX)

**S1 Table. List of UK Biobank phenotypes analyzed in the study.** List of UK Biobank phenotypes analyzed in the study. Phenotype name, the source field ID in UK Biobank (UKB Field ID), phenotype ID in Global Biobank Engine (GBE ID), the number of individuals (N) are shown.
(XLSX)

**S2 Table. Consistency of the genotype calls for four protein-altering variants in *ANGPTL7* between genotyping array and exome sequencing data.** Consistency of the genotype calls for four protein-altering variants in *ANGPTL7* between genotyping array and exome sequencing data. Variant including chromosome, position, reference, and alternate allele (hg19), the rs identifier of the genetic variant (rsID), amino acid nomenclature (HGVSp), genotype call from the array (Array) and exome data (Exome), and the number of individuals (N). Inconsistent variant calls are highlighted in bold font.
(XLSX)

**S3 Table. Number of individuals stratified by genotype of rare (0.01% < MAF < 1%) protein-altering variants in *ANGPTL7*.** Number of individuals stratified by genotype of rare (0.01% < MAF < 1%) protein-altering variants in *ANGPTL7*. The combination of genotypes is shown in the first four columns (rs200058074, rs28991002, rs28991009, and rs143435072) as well as the number of British individuals with the genotype combination in UK Biobank (n_observed). The expected number of individuals is computed under the Hardy-Weinberg equilibrium model and the independence assumption (n_expected, Method).
(XLSX)

**S4 Table. GCTA estimates of phenotypic variance explained by the 4 rare variants in *ANGPTL7* for the IOP measures and glaucoma.** GCTA estimates of phenotypic variance

explained by the 4 rare variants in *ANGPTL7* for the IOP measures and glaucoma. The phenotype (Phenotype and GBE_ID), the estimated local heritability (V(G)/Vp), standard error (SE), and p-value (*P*) are shown. The SE and *P* are estimated based on from Jackknife method.
(XLSX)

**S5 Table. *ANGPTL7* IOP protein-altering variant association for 6 IOP measurements in UK Biobank.** *ANGPTL7* IOP protein-altering variant association for 6 IOP measurements in UK Biobank. The association statistics for 6 IOP traits (corneal compensated IOP [median INI2005254, right: INI5254, and left: INI5262] and Goldman-correlated IOP [median INI2005255, right: INI5255, and left: INI5263]) are shown. The phenotype (GBE_ID), variant (chromosome, position, reference, and alternate allele [hg19]), rsID, the HGVS amino acid nomenclature (HGVSp), and the estimated regression coefficient with 95% confidence intervals (BETA [95% CI]), and p-value of association (*P*) are shown.
(XLSX)

**S6 Table. *ANGPTL7* IOP protein-altering variant association for 6 IOP measurements in UK Biobank.** Pairwise genetic correlation of IOP phenotypes. The genetic correlation (rg) for pairs of traits (Trait 1 and Trait 2, shown as GBE ID for 6 IOP traits (corneal compensated IOP [median INI2005254, right: INI5254, and left: INI5262] and Goldman-correlated IOP [median INI2005255, right: INI5255, and left: INI5263]) is shown with the standard error estimates (SE) based on Jackknife.
(XLSX)

**S7 Table. Glaucoma definitions in FinnGen.** Glaucoma definitions in FinnGen. ICD-codes are used in the Finnish hospital discharge and cause-of-death registries. ATC-codes are used in the Social Insurance Institution prescription drug purchase registry. All endpoint definitions in the FinnGen phenome-wide association analysis are available online (https://www.finngen.fi/fi/node/68).
(XLSX)

**S8 Table. FinnGen summary of *MYOC* p.Gln168Ter and *ANGPTL7* p.Arg220Cys carriers in primary open glaucoma and all glaucoma cases.** FinnGen summary of *MYOC* p. Gln168Ter and *ANGPTL7* p.Arg220Cys carriers in primary open glaucoma cases (A) and all glaucoma (B). The numbers of primary open-angle glaucoma (POAG) cases/controls stratified by genotype are shown. FinnGen summary of genotype counts for *ANGPTL7* p.Arg220Cys in unrelated individuals in primary open glaucoma cases (C) and all glaucoma (D).
(XLSX)

**S9 Table. *ANGPTL7* protein-altering variant association with reproductive fitness.** *ANGPTL7* protein-altering variant association with reproductive fitness, (A) the number of live births and (B) the number of children fathered. Variant includes chromosome, position, reference, and alternate allele (hg19). rsID—the rs identifier of the genetic variant. HGVSp— the HGVS protein sequence name. MAF—the minor allele frequency in UK Biobank British population. Beta—estimated regression coefficient with 95% confidence intervals. P—p-value of association.
(XLSX)

**S10 Table. The PheWAS results for the four protein-altering variants in *ANGPTL7* in UK Biobank.** The PheWAS results for the four protein-altering variants in *ANGPTL7* in UK Biobank. The association summary statistics with P < 1.0 x $10^{-3}$ and SE < .5 are shown.
(XLSX)

## Acknowledgments

This research has been conducted using the UK Biobank Resource under Application Number 24983, "Generating effective therapeutic hypotheses from genomic and hospital linkage data" (http://www.ukbiobank.ac.uk/wp-content/uploads/2017/06/24983-Dr-Manuel-Rivas.pdf). Based on the information provided in Protocol 44532 the Stanford IRB has determined that the research does not involve human subjects as defined in 45 CFR 46.102(f) or 21 CFR 50.3 (g). All participants of UK Biobank provided written informed consent (more information is available at https://www.ukbiobank.ac.uk/2018/02/gdpr/). For the Finnish Institute of Health and Welfare (THL) driven FinnGen preparatory project (here called FinnGen), all patients and control subjects had provided informed consent for biobank research, based on the Finnish Biobank Act. Alternatively, older cohorts were based on study specific consents and later transferred to the THL Biobank after approval by Valvira, the National Supervisory Authority for Welfare and Health. Recruitment protocols followed the biobank protocols approved by Valvira. The Ethical Review Board of the Hospital District of Helsinki and Uusimaa approved the FinnGen study protocol Nr HUS/990/2017. The FinnGen preparatory project is approved by THL, approval numbers THL/2031/6.02.00/2017, amendments THL/341/6.02.00/2018, THL/2222/6.02.00/2018 and THL/283/6.02.00/2019. All DNA samples and data in this study were pseudonymized. We thank all the participants in the UK Biobank and Finnish Biobanks used in FinnGen studies. We thank Sirpa Soini, Fatima Rodriguez and David Amar for invaluable feedback on the manuscript.

## Author Contributions

**Conceptualization:** Mark J. Daly, Manuel A. Rivas.

**Data curation:** Yosuke Tanigawa, Juha Karjalainen, Tuomo Kiiskinen, Mark J. Daly, Manuel A. Rivas.

**Formal analysis:** Yosuke Tanigawa, Juha Karjalainen, Tuomo Kiiskinen, Guhan Venkataraman, Susanna Lemmelä, Aki S. Havulinna, Mark J. Daly, Manuel A. Rivas.

**Funding acquisition:** Mark J. Daly, Manuel A. Rivas.

**Investigation:** Yosuke Tanigawa, Juha Karjalainen, Mark J. Daly, Manuel A. Rivas.

**Methodology:** Yosuke Tanigawa, Guhan Venkataraman, Mark J. Daly, Manuel A. Rivas.

**Project administration:** Aarno Palotie, Mark J. Daly, Manuel A. Rivas.

**Resources:** Mark J. Daly, Manuel A. Rivas.

**Software:** Yosuke Tanigawa, Guhan Venkataraman, Mark J. Daly, Manuel A. Rivas.

**Supervision:** Mark J. Daly, Manuel A. Rivas.

**Visualization:** Yosuke Tanigawa, Mark J. Daly, Manuel A. Rivas.

**Writing – original draft:** Yosuke Tanigawa, Mark J. Daly, Manuel A. Rivas.

**Writing – review & editing:** Yosuke Tanigawa, Michael Wainberg, Juha Karjalainen, Tuomo Kiiskinen, Susanna Lemmelä, Joni A. Turunen, Robert R. Graham, Aki S. Havulinna, Markus Perola, Aarno Palotie, Mark J. Daly, Manuel A. Rivas.

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
