## [Decision Letter · Decision Letter 0]

15 Oct 2019

Dear Dr Rivas,

Thank you very much for submitting your Research Article entitled 'Rare protein-altering variants in ANGPTL7 lower intraocular pressure and protect against glaucoma' to PLOS Genetics. Your manuscript was fully evaluated at the editorial level and by independent peer reviewers. The reviewers appreciated the attention to an important problem, but raised some substantial concerns about the current manuscript. Based on the reviews, we will not be able to accept this version of the manuscript, but we would be willing to review again a much-revised version. We cannot, of course, promise publication at that time.

If you decide to revise the manuscript for further consideration at PLOS Genetics, please aim to resubmit within the next 60 days, unless it will take extra time to address the concerns of the reviewers, in which case we would appreciate an expected resubmission date by email to plosgenetics@plos.org.

[LINK]

We are sorry that we cannot be more positive about your manuscript at this stage. Please do not hesitate to contact us if you have any concerns or questions.

Yours sincerely,

Jessica N. Cooke Bailey, Ph.D.

Guest Editor

PLOS Genetics

Scott Williams

Section Editor: Natural Variation

PLOS Genetics

While this manuscript has potential to be high-impact, there are major flaws cited by the reviewers which must be addressed.

Reviewer's Responses to Questions

**Comments to the Authors:**

Reviewer #1: Primary open-angle glaucoma (POAG), and its endophenotypes, including intraocular pressure (IOP), are strongly heritable, and numerous risk loci have been identified through genomwide association studies (GWAS). The role of rare genetic variation, on the other hand, has received far less attention. This manuscript addresses the role of rare variants in IOP using two large cohorts, the UK Biobank and FinnGen. A single-variant association analysis of the UK Biobank sample revealed several nonsynonymous coding variants in ANGPTL7 that, on average, lower IOP. An independent missense variant in ANGPTL7 the FinnGen sample also reduces IOP, suggesting a general protective role of ANGPTL7 inactivation in POAG.

This study is of high potential impact, considering the large public health burden of POAG in the elderly and the paucity of knowledge of the contribution of rare genetic variation. A strong point of this article is the large sample size, suitable for studies of rare variants, and meticulous quality control and documentation of the analysis on the UK Biobank cohort. The manuscript is written fairly well, with a few typos and minor grammatical errors.

However, the manuscript is marred by improperly conducted statistical analyses and lack of important information in the Methods and Discussion. Specifically, a meta-analysis is inappropriate for measuring the aggregate effect of multiple genetic variants. A simple burden test would be more appropriate, as well as an estimate of the proportion of phenotypic variation explained by the variants. If the authors feel that a meta-analysis is suitable here, they need to explain their rationale very carefully. Moreover, an analysis of genetic correlation produced an impossible correlation estimate of 1.08, suggesting that some other quantity besides correlation is being measured. These serious errors cast doubt on the reliability of the main findings, even though the primary association analysis seems sound. It is surprising that a genomewide gene-level association analysis of rare variants (e.g., burden test or SKAT), which would increase power to detect rare-variant aggregate effects, was not conducted. Methods for several analyses are missing, including the measurement of genetic correlation (Suppl. Fig 7) and the PheWAS (Suppl. Fig. 8). The PheWAS is only mentioned once in the main text, and only for citing the association result for Glaucoma; the significance of the other results from the PheWAS are not discussed.

The Discussion is perfunctory and is lacking important content. What are the limitations of the study? Specifically, how does the lack of overlap between POAG risk loci and IOP loci (e.g., Springelkamp et al., 2017, PMID 28073927) affect the significance of these results for treating POAG? What are the implications of the differences in allele frequency of IOP-lowering variants in the Finnish and UK populations? How do the new findings for ANGPTL7 fit in with what is already known about its role in POAG and IOP?

Specific comments:

1. Introduction, p. 2: Khawaja et al. (ref. 6) alone identified 68 risk loci for IOP. The total number of risk loci, including previous studies, including is larger. Choquet et al (2018) PMID 29235454 and Hysi et al. (2014) PMID 25173106 should also be cited, The term “unequivocally implicated” should be made clearer: does this mean replicated within one study? between two or more independent studies?

2. Results, p. 3: How was the joint association analysis for the three less significant ANGPTL7 SNPs performed? By a burden test?

3. Results p. 4: The lack of significance in the associations with glaucoma may also be explained by misclassification in the glaucoma phenotype on account of its being based on EHR, and by the likely presence of normal-tension POAG patients within the glaucoma cases.

4. Results, pp. 5-6. The last paragraph of the Results belongs in the Discussion, except for the sentence on tissue-specific expression of ANGPTL7.

5. Discussion: Does the Finnish population have a different prevalence of POAG than the UK?

6. Methods: Was the entire age range of the UK Biobank dataset included? The genetic determinants of IOP before age 40 may well be different than in older individuals. Was the average age of rare-variant carrying individuals much different from that of the entire sample?

7. Methods, p. 8 top: What is the “Array” predictor in the logistic regression model?

8. Suppl. Fig. S1: Considering the very large sample size, a density plot comparing IOP in cases and controls will provide more information than a boxplot. See Fig. 4 of Martin et al. (2017) PMID 28366442 for an example of overlapping density distributions.

9. Suppl. Fig. S7: This information would be much more concisely shown in a table, or even in the text. It is not clear what the correlations are between: the three x-axis labels each mention only one variable.

10. Suppl. Fig. S8: This PheWAS analysis isn’t mentioned anywhere in the text, except to indicate that the association of R220C with glaucoma was highly significant. Can it be omitted from the paper except for the glaucoma-related phenotypes?

11. Suppl. Table S1: The data would be more straightforward to interpret in the form of a small table for each variant with counts for each genotype pair observed (including NA), rather than one large table with the counts in a single column.

12. Suppl. Table S6: This appears to be raw, unformatted output, and should be formatted as a table.

Reviewer #2: This is an interesting study examining rare variant associations with glaucoma and its major endophenotype, IOP. There are several issue that need addressing.

Major comments:

- It is an odd approach to take Goldmann-correlated IOP of the right eye as a primary measure, and then not display results for the left eye measure or the corneal-compensated measures (only show genetic correlations). Are the authors hypothesizing that genetic associations with IOP may only influence one eye and not the author? If not, a better approach is to include both eyes and adjust for the correlation using a random-effects approach, or to simply take the mean of right and left eye measures.

- Why is the primary analysis for Goldmann-correlated IOP? Corneal compensated IOP has been shown to be more reflective of true physiological IOP, and less influenced by corneal artefact. Could ANGPTL7 variation actually be influencing the cornea rather than IOP?

- Were the IOP variables cleaned prior to analysis? If so, how?

- What does the meta-analyzed effect estimate mean when combining effects at multiple different variants (Supp Tables 6 and 9)? Is this the effect you would expect to see if someone had all these variants together? It seems odd to me that you would search for IOP-lowering variants in a gene, and then meta-analyze these selected variant effects together. Surely this is biased and misleading? Unless the authors can make a very strong rationale, I would remove sections on "combined significance".

- The definition of glaucoma in UK Biobank , a major outcome variable in the paper, is not clear. How many were identified using self-report? How many by hospital episode statistics? Why did the authors not limit to POAG HES codes? How were controls defined, given that the glaucoma question was not administered to the whole cohort? Given this is a key outcome variable, I would recommend the authors present a flow chart for derivation of glaucoma status as well as IOP.

- Is it possible that the protein alteration increases function of the gene? What evidence do the authors have that the functional consequence of the identified variants is reduced gene function? Unless strong, the authors should temper the strength of the language they use to describe the effect.

- The discussion is disappointingly short. How does this finding sit with other genetic discoveries for IOP and glaucoma? How does this fit in with what is known about IOP-related anatomy and physiology? What type of treatments might target the gene or its downstream effects, and how would the drug be delivered? Is there a plausible explanation for the hypothesis that the authors suggest regarding modifying the glaucoma risk of patients with MYOC mutations?

Minor comments:

- IOP is not the sole predictive factor for glaucoma

- The statement that there are "total of 68 independent loci have been unequivocally implicated in glaucoma" seems incorrect - the papers the authors cite do not reflect this on deeper reading.

- The text in the 2nd paragraph regarding "signals were consistently observed in left eye IOP measure" is not clear. Were previous analyses only carried on right eyes (if so, this is not clearly stated in the Results)? Are they referring to Goldmann-correlated IOP here? Results should be presented more robustly. Anyway, the authors may change their analytical approach based on the above.

Reviewer #3: This is a well-written paper describes several rare ANGPTL7 protein-coding variants that are associated with lower intraocular pressure (IOP) in participants from the UK Biobank and associated with decreased risk of glaucoma in the FinnGen dataset. Several points to address:

1) The overall beta for intraocular pressure reduction by heterozygous variants is very small and even the homozygous Gln175His would not be expected be within the resolution of clinical measurement or to be clinically relevant. This should be discussed especially in regard to therapeutic development.

2) The authors note that an ANGPTL7 rare variant is likely responsible for the 1p36 signal reported in Khawaja et al. It would be interesting to note if this signal has been observed in other IOP GWAS such as Choquet et al., 2018.

3) While overall the examination of the ANGPTL7 effects on MYOC368ter cases is interesting there are several questions about this result. First, since the FinnGen glaucoma cases are not actually examined, but defined by ICD codes, its possible that some of the MYOC 368ter 'noncases' are actually cases- this is particularly relevant when considering a recent study that has shown that some patients with MYOC 368ter can have glaucoma without intraocular pressure elevation (Fingert et al., JAMA Ophthalmology). Second, was the distribution of ANGPTL7 variant carriers among MYOC 368ter carriers statistically significant?

4) A limitation of the study is that all the glaucoma cases are defined by ICD codes without any clinical validation. These codes used to define case-control status also include 'glaucoma secondary to eye trauma', 'secondary to eye infection or other eye disorders' and 'secondary to drugs'. Eye traumas are not genetic, while drugs causing glaucoma are primarily corticosteroids, which could drive these results considering the potential role of ANGPTL7 in steroid-responsive glaucoma (see point 6 below). Moreover, including all types of glaucoma is concerning as various forms of glaucoma have very different mechanisms and some can be difficult to distinguish without expert evaluation. Given the very high prevalence of exfoliation glaucoma in Finland this would be of special concern in the FinnGen population. Further replication of these findings in a cohort of individuals diagnosed by clinical experts would be helpful.

5) As ANGPTL7 has been shown to be increased in glaucoma secondary to steroid (glucocorticoid) exposure and this type of glaucoma has a specific ICD code is it possible to examine this subgroup among the UK Biobank cases? Even better in patients who have been clinically diagnosed to have this type of glaucoma by glaucoma experts? Is it possible that this subgroup is driving the UKBiobank results? Showing that these variants are protective in POAG patients examined by a clinical expert with knowledge of the history of steroid exposure in the patient would also be helpful.

6) There is very little discussion of any potential protective role for the ANGPTL7 protein or functionally how loss of function variants could impact intraocular pressure and glaucoma. Additionally, while the nonsense variant is likely to be loss of function, this may not actually be the case as the most common MYOC variant (368ter) is actually a gain of function. Its not clear if the missense alleles are loss or gain of function. Again, similar to MYOC the missense alleles are gain of function. This information is very relevant to the development of ANGPTL7 based therapies.

7) This sentence is confusing, “Given these findings, we next asked whether any of these putative IOP-lowering genetic variants showed effects consistent with reducing glaucoma risk in an independent set of unrelated British individuals that do not have IOP measures (4,269 cases and 251,355 controls).” Are these glaucoma cases not included in the set of UK Biobank individuals with eye phenotype data?

**Have all data underlying the figures and results presented in the manuscript been provided?**

Reviewer #1: Yes

Reviewer #2: Yes

Reviewer #3: Yes

PLOS authors have the option to publish the peer review history of their article (what does this mean?). If published, this will include your full peer review and any attached files.

Reviewer #1: No

Reviewer #2: No

Reviewer #3: No

---

## [Decision Letter · Decision Letter 1]

18 Feb 2020

Dear Dr Rivas,

We are pleased to inform you that your manuscript entitled "Rare protein-altering variants in ANGPTL7 lower intraocular pressure and protect against glaucoma" has been editorially accepted for publication in PLOS Genetics. Congratulations!

Yours sincerely,

Jessica N. Cooke Bailey, Ph.D.

Guest Editor

PLOS Genetics

Scott Williams

Section Editor: Natural Variation

PLOS Genetics

Comments from the reviewers (if applicable):

Reviewer's Responses to Questions

**Comments to the Authors:**

Reviewer #1: The manuscript has been greatly improved and is acceptable for publication.

Reviewer #2: The revision has adequately addressed my concerns.

Reviewer #3: In this revised manuscript and response to reviewers the authors have addressed the concerns raised by the review.

**Have all data underlying the figures and results presented in the manuscript been provided?**

Reviewer #1: Yes

Reviewer #2: Yes

Reviewer #3: Yes

PLOS authors have the option to publish the peer review history of their article (what does this mean?). If published, this will include your full peer review and any attached files.

Reviewer #1: No

Reviewer #2: No

Reviewer #3: No

**Data Deposition**

http://datadryad.org/submit?journalID=pgenetics&manu=PGENETICS-D-19-01516R1

**Press Queries**

---

## [Editor Report · Acceptance letter]

17 Apr 2020

PGENETICS-D-19-01516R1 

Rare protein-altering variants in ANGPTL7 lower intraocular pressure and protect against glaucoma 

Dear Dr Rivas, 

We are pleased to inform you that your manuscript entitled "Rare protein-altering variants in ANGPTL7 lower intraocular pressure and protect against glaucoma" has been formally accepted for publication in PLOS Genetics! Your manuscript is now with our production department and you will be notified of the publication date in due course.

With kind regards,

Kaitlin Butler

PLOS Genetics

On behalf of:
